# Nanopatterned Polymer Molds Using Anodized Aluminum Templates for Anti-Reflective Coatings

**DOI:** 10.3390/polym13193333

**Published:** 2021-09-29

**Authors:** Soon Hyuk Lim, Nguyễn Hoàng Ly, Jung A. Lee, Ji Eun Kim, Se-Woong La, Vu Thi Huong, Thi-Giang Tran, Ngoc Thanh Ho, Seung Man Noh, Sang Jun Son, Sang-Woo Joo

**Affiliations:** 1Department of Chemistry, Gachon University, Seongnam 13120, Korea; 2952305@naver.com (S.H.L.); nguyenhoangly2007@gmail.com (N.H.L.); sb52431@naver.com (J.A.L.); wnpwnp81@naver.com (J.E.K.); 2Department of Chemistry, Soongsil University, Seoul 06978, Korea; sktpdnd9668@daum.net (S.-W.L.); huongvu172018@gmail.com (V.T.H.); trangianghup@gmail.com (T.-G.T.); ngocthanhkd94@gmail.com (N.T.H.); 3Research Center for Green Fine Chemicals, Korea Research Institute of Chemical Technology, Ulsan 44412, Korea

**Keywords:** nanocone pattern, transparent coating films, anodized aluminum oxide, anti-reflective surfaces, moth-eye structures, shape-controlled fabrication

## Abstract

This work introduces a facile geometry-controlled method for the fabrication of embossed and engraved polymeric moth-eye-inspired nanostructures in imprinting molds using anodic aluminum oxide (AAO) templates, resulting in a novel anti-reflective transparent coating. The moth-eye nanostructures are prepared directly on the surface of a flexible polyethylene terephthalate (PET) substrate. As a prerequisite procedure, a UV-curable polyurethane acrylate resin is spun on the PET. The shape of the moth-eye nanostructures can then be adjusted by controlling the size and shape of the nanopores in the AAO templates. Both embossed and concaved polymer moth-eye nanostructures were successfully mounted on a PET substrate. Embossed polymer replica molds were prepared using the AAO master templates in combination with an imprinting process. As revealed by field-emission electron microscope (FE-SEM) images, conical nanopatterns in the AAO template with a diameter of ~90 nm and a depth of ~100 nm, create a homogeneous embossed morphology in the polymer moth-eye nanostructure. The polymeric molds with the depths of 300 and 500 nm revealed the amalgamated structures in their apexes. In addition, a dip-imprinting process of the polymeric layers was implemented to yield a concaved mold by assembly on the surface of the 100 nm embossed polymer mold substrate. Considering that the embossed structures may be crumbled due to their protuberant shapes, the concaved geometries can have an advantage of stability in a certain application concerning physical degradation along with a higher transmission by ~2%, despite somewhat nonuniform structure. The experimental and theoretical results of this study indicate that this polymer layer has the potential for use in anti-reflective coating applications in transparent films.

## 1. Introduction

Nanostructures have been used for various applications in chemistry and materials science [1]. Nanotemplates have recently emerged as potential platforms for the fabrication of functional materials with many uses in multiple fields, including transparent coatings [2] and solar cells [3]. The bionic prototypes and antireflection principles inspired by nature are discussed in the recent review [4]. Hexagonal nipple-array of subwavelength conical nanostructures have a diameter of ca. 100 nm and can provide broadband anti-reflectivity to enhance photon collection capability [5]. The shapes and sizes appeared to affect reflectivity performance [6,7,8]. Several nanostructures have been developed for efficiently hydrophobic and anti-reflective coating materials [9,10,11]. To obtain the anti-reflection effects of moth-eye-like nanostructures, a high- density array with homogeneous morphology are prerequisites for practical applications.

Recent studies have demonstrated several methodologies to fabricate moth-eye nanostructures including the Langmuir–Blodgett technique [12], lithography [9,13,14], imprinting processes [15], double replication of a silicon master [10], dip-coating solid silica nanoparticles [11], and a combination of ultrasonic vibration demolding and precision hot embossing [16]. In particular, there has been a series of studies on the AAO template-assisted synthesis of moth-eye-inspired nanostructures reporting significant properties such as superhydrophobicity [17], anti-reflectivity [18,19], and ultrasensitive substrates for surface-enhanced Raman scattering-based detection [4,20]. Such templates represent a low-cost and facile process to provide an alternative to the above-mentioned techniques.

Polymer-based materials have attracted considerable attention due to their flexibility, hydrophobicity, and universal applicability [21,22,23,24,25]. Several studies have been conducted around the development of novel polymer layers, including poly(methyl methacrylate) coatings to enhance the light amplification of perovskite films [26] and superamphiphobic properties of silica-fluoropolymer coating layer [27]. In addition, near-surface mounting methods have been investigated in relation to fiber reinforced polymer strengthening [21,28] and reinforcing concrete bridges [23,29]. These studies have demonstrated the importance of homogeneous morphology in polymeric patterns. Based on the previous routine procedures, there is a practical need to develop the fabrication of various polymer-based moth-eye nanostructures, through direct imprinting using geometry-controlled AAO templates in building master molds for patterned polymer materials. Although the preparation of moth-eye nanostructures with noble metal materials and AAO platforms has been widely investigated, the novel technique of fabricating concave polymer-based nanostructures requires further research to implement this into practical applications.

In this work, an anti-reflective film of polymeric moth-eye nanostructures was assembled on the PET surface, by removing the AAO template. This flexible nanostructured film, with a thickness of c.a. 100 nm, exhibited not only the superhydrophobic properties but also high transmission in the visible and near-infrared region. AAO is a versatile nanostructure material due to its economical and reproducible qualities. Since AAO platforms have been successfully employed in preparing nanostructured metal materials with high uniformity in numerous shapes, including wires, tubules, rods, and dots, the present method of combining an AAO template with the UV imprinting process appears to be a convenient, straightforward, low-cost, and reusable method to build moth-eye-inspired polymeric nanostructures with a promise for anti-reflective coating applications.

## 2. Materials and Methods

### 2.1. Materials

Aluminum foil (99.99%) was obtained from Alfa Aesar (Karlsruhe, Germany). Perchloric acid (70%), ethyl alcohol anhydrous (99.9%), oxalic acid (99.5%), chromium oxide (Ⅳ) (99%), phosphoric acid (99.5%), and acetone (99.5%) were purchased from Daejung Chemicals (Siheung, Korea). The fluorine hydrophobic demolding reagent, UV-curing polyurethane acrylate resins, and additives were synthesized by referring to the recent report [30].

### 2.2. Preparation and Surface Modification of Moth-Eye-Inspired Polymer Nanostructures

Annealed aluminum foil was degreased using a Branson 8510 sonicator (Brookfield, CT, USA) in acetone for 1 h and then electro-polished with a mixture of perchloric acid and ethanol (1:5, *v*/*v*) for 6 min 30 s at 0 °C and 15 V. These templates were first-anodized in 0.3 M oxalic acid solution under the condition of 10 °C and 40 V for 8 h. The nonuniform aluminum oxide membrane that resulted from the first-anodization was performed by a mixture solution of chromium oxide (1.8 wt.%) and phosphoric acid (6 wt.%) at 60 °C for 5 h. The depths and pore diameters in nanocone-shape AAO molds could be controlled by varying the 2nd anodization and pore widening times from 17–83 s and 3–5 min, respectively. To obtain a conical nanopore of ~100 nm in depth and ~90 nm in diameter, the templates were secondly anodized under the same conditions as the first-anodization for 15 s. The templates were then immersed in an aqueous solution of phosphoric acid (5 wt.%) under the condition of 30 °C for 5 min. The processes of the second anodization and pore-widening were repeated six times.

Next, 10 drops of the fluorine-containing surface-treatment solution were applied to the bottom of either a beaker or glass petri dish, and the AAO templates with conical shape nanopores were placed on the top for an efficient demolding. In order to evaporate the solvent on the surface of the template, the samples were placed in an oven at 130 °C for 30 min. They were then allowed to cool at room temperature and washed twice with ethanol. Following this surface modification of the AAO mold with the polymer solution, the template was placed on the resin on the PET film and rolled to fabricate a polymer layer as thin as possible, which was then cured by the UV irradiation through the upward facing PET substrate for 30 s. The AAO template was then removed to reveal a polymer layer on the PET replicating the moth-eye nanopatterns of the template. This was cured with UV for another 30 s.

Figure 1 illustrates the fabrication process of a moth-eye nanostructure in polymer resin on a PET substrate using an AAO template as an imprinting mold. In the first step, a high-purity aluminum foil was anodized in a two-stage process leading to conical nanopore array inside the AAO template. Each nanocone was specifically designed with a 90 nm diameter and a 100 nm depth. Subsequently, the AAO platform was reacted with a polymer to obtain a lower surface energy than the original material. The AAO surface could deform the nanostructure during the template removal and the UV curing. The shape and size of the nanopore array inside the AAO template depended on the conditions of each anodization phase. This method was performed with the second anodization under the same conditions as the first for 15 s. The morphology characterization of the AAO templates was conducted by FE-SEM. This soft molding technique combined with UV imprinting process exhibited an efficient fabrication of flexible nanostructures [31]. The method not only produces patterns with the advantage of continuous production over areas as large as several centimeters but is also a particularly high-speed manufacturing process. The combination of UV imprinting and demolding to form a transparent film of polymeric moth-eye nanostructures on a PET substrate is illustrated in Figure 1.

### 2.3. Physicochemical Characterization and COMSOL Multiphysics Simulation

AAO templates and moth-eye-like nanostructures of polymer layers were examined via a JEOL JSM-7500F FE-SEM (Tokyo, Japan). Contact angle images were obtained by a SEO Phoenix-10 analyzer (Suwon, Korea). UV-curing was performed with a Lightzen 60 PH portable device. (Gunpo, Korea). The optical transmission measurements were conducted using a Shimadzu 2450 spectrophotometer (Tokyo, Japan). A COMSOL simulation (Burlington, MA, USA) was performed with a modification for an organic material by referring to the previous work on the semiconductor materials [8]. The absorption and scattering spectra of polymer-based moth-eye-like nanostructure were calculated using the package of COMSOL Multiphysics Version 5.3a. Each nanocone had a 90 nm diameter and a 100 nm (300 nm, 500 nm) depth with a refractive index of 1.5. This package was also used to simulate the absorption and scattering spectra of the moth eye structure (MES) with the incident light. The geometric parameters of the nanostructure were realized by the ‘parameter’ function in ‘global definition’, which could easily change the parameter settings of the entire model. The single hexagonal structure of MES with the 6 tips was chosen to represent the MES array. The length and the tip radius were 100 nm and 45 nm, respectively with the substrate thickness of 100 nm. This model assumed that the real part of the refractive index of air was 1 and the imaginary part of the refractive index was 0. The refractive index of MES was set to be 1.5. The length of the MES substrate was introduced into the middle of an air spherical domain (φ = 2 μm) to avoid reflection by a perfectly matched layer boundary condition. The incident light wavelength range of this simulation model was 400 nm to 2000 nm; thus, the maximum cell size is 65 nm, and the minimum cell size was 2 nm.

## 3. Results and Discussion

For UV imprinting, coating the PET with a polymer is a prerequisite procedure, and the nanostructured moth-eye layer was applied using an AAO master mold, which was subsequently compressed against polymer-coated PET film using a roll bar. The sample was cured with the UV light for 30 s and again for 30 s, once the AAO template had been removed, thus generating a replica moth-eye nanostructure. FE-SEM images of the AAO master mold templates exhibited well-controlled nanocone arrays. The shapes and sizes of the imprinted polymeric moth-eye-like nanostructure arrays were found to depend on the structures of nanopores arrays in AAO template molds. According to the FE-SEM images, only a ~100 nm structure was found to be eligible to produce a stable embossed polymer mold shape as demonstrated in Figure 2. The neighboring apexes in nanocones appeared to be amalgamated for the 300 and 500 nm templates in depth, presumably due to their strong van der Waals interactions.

Figure 3a,b shows the top-view FE-SEM images of numerous embossed conical shaped polymeric nanostructures prepared using the AAO template with the diameter and depth of ~90 nm and ~100 nm based on the results in Figure 2. FE-SEM images indicated that a master mold exhibited a homogeneous morphology. To demonstrate a property of the self-cleaning and self-curing behavior of a moth-eye-like nanostructure, the surface of AAO template was coated by the surface-treatment agents. Figure 3 illustrates FE-SEM images of the polymeric moth-eye nanostructures prepared with the AAO templates. The homogenous morphology of the surface layers is a result of controlling the nanopore array on the PET film with each nanocone measuring ~90 nm and ~100 nm in diameter and length, respectively, and ~105 nm between the two nanocones. Figure 3c,d revealed the concaved polymeric nanostructures using the embossed polymer replica. As indicated from the collapsed structured in Figure 2d,f, the embossed structures may have disadvantages of the possible collapse and amalgamation with the neighboring nanocone structures. Despite somewhat nonuniform shapes, the concaved polymer molds from the embossed replica may be suitable for a certain type of anti-reflective coating.

Photos of the embossed and concaved anti-reflective polymeric coating structures on the PET substrates with a dimension of 3 cm × 5 cm are shown in Figure 4a,b, respectively. AAO surfaces were prepared by anodization to exhibit hydrophobicity, followed by the surface-treatment coating. A water contact angle measurement of the AAO surfaces was estimated that the corner angles of conical-shape structures to be 127°. The current water contact angle measurements of the embossed and concaved polymer surfaces were estimated that the corner angles of conical-shape polymeric structures were 132.5 (±2.7)° and 128.2 (±0.9)°, respectively (Figure 4c,d, Table 1), with a slight increase from the AAO substrates. Before and after the treatment of the surfaces of the AAO templates, the fluorine-containing surface-treatment polymer had to be introduced to make the polymeric molds superhydrophobic.

Polymeric moth-eye nanostructures have been found to exhibit anti-reflective properties according to previous works [32,33]. In the present experiments, AAO molds were used as imprinting templates to produce polymeric moth-eye nanostructures. UV-curing was applied after the imprinting and the removal of the template. These nanostructure layers can enable efficient practical applications due to their properties including hydrophobicity, flexibility, and transparency. To examine the anti-reflective properties, the absorption and scattering spectra of polymeric moth-eye-like nanostructure were computed using the COMSOL multiphysics software. The results of the simulated spectra of embossed and concaved polymer-based moth-eye-like nanostructures as depicted in Figure 5a,b, respectively, which indicates a negligible absorption, which is verified by almost perfect transmission at 450–800 nm, as shown in the experimental measurements of Figure 5c,d. The measured transmission values were found to range from 93.4% to 94.2% for the embossed polymer molds, whereas they varied from 95.8% to 96.9% for the concaved molds in the wavelength region between 500 and 800 nm (Table 2 and Table 3). Notably, the concaved molds showed approximately 2% higher transmittance values than those of the embossed ones, presumably due to different scattering properties. The concaved mold was found to have a higher scattering cross section in the presented COMSOL calculations. Although not shown here, both the 300 and the 500 nm conical structure exhibited imperceptible calculated values as that of 100 nm, along with the insignificant absorption above 800 nm to 2000 nm. Considering that the coating materials in the present study are supposed to be applied in outdoor sensor devices such as a lidar sensor, the high transmission and anti-reflective properties should be suitable for the visible and near-infrared region.

## 4. Conclusions

A moth-eye nanostructured polyurethane acrylate mold was successfully fabricated on a PET substrate using a direct UV imprinting process with the AAO template containing conical shapes. The anti-reflective coating properties of the polymeric layer were evaluated using contact angle measurements, UV-Vis-NIR transmittance, and computer simulations. The following aspects can be summarized from the present study.

Both embossed and engraved nanopatterned polymeric mold structures were tested for potential applications in transparent coating.Only an AAO master mold in the depth of 100 nm is found to be eligible to produce a stable embossed polymer mold shape.The neighboring apexes in nanocones appeared to be amalgamated for the 300 and 500 nm templates in depth.Contact angle measurements of the embossed and concaved polymer surfaces were estimated to be 132.5 (±2.7)° and 128.2 (±0.9)°, respectively.Concaved molds showed approximately 2% higher transmittance values than those of the embossed ones.

The developed nanostructures showed promise for real applications due to their anti-reflection and hydrophobicity. This fabrication method provides a potential technology for designing multifunctional substrates and building moth-eye-inspired materials. The anti-reflective layer not only has wide universality but also suggests numerous potential applications in many fields, including flexible display screens, transparent coatings, hydrophobic films, and anti-dazzle glasses.

## Figures and Tables

**Figure 1 polymers-13-03333-f001:**
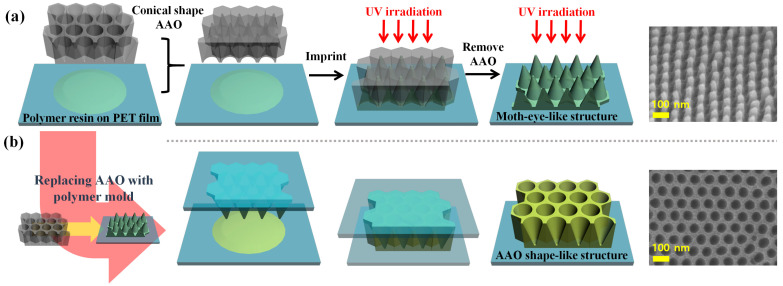
Schemes of UV-curable polyurethane acrylate moth-eye-like nanostructures on PET substrates for (**a**) embossed and (**b**) concaved polymer molds.

**Figure 2 polymers-13-03333-f002:**
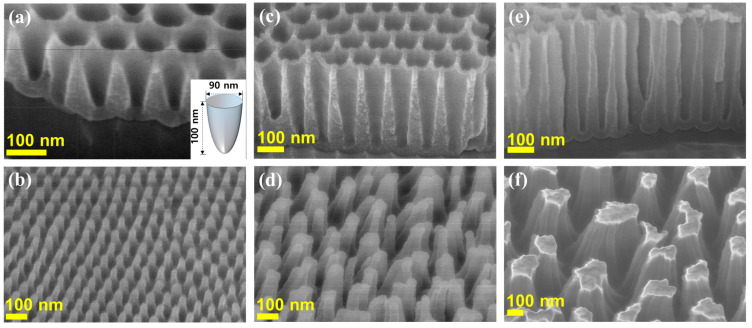
FE-SEM images of the nanocone AAO templates and the resulting moth-eye-like nanostructure from UV-curable polyacrylate urethane resins on a PET substrate with the depths of (**a**,**b**) ~100, (**c**,**d**) ~300, and (**e**,**f**) ~500 nm. All the scale bars are 100 nm. Inserted picture illustrates one nanocone of ~90 nm in diameter and ~100 nm in length.

**Figure 3 polymers-13-03333-f003:**
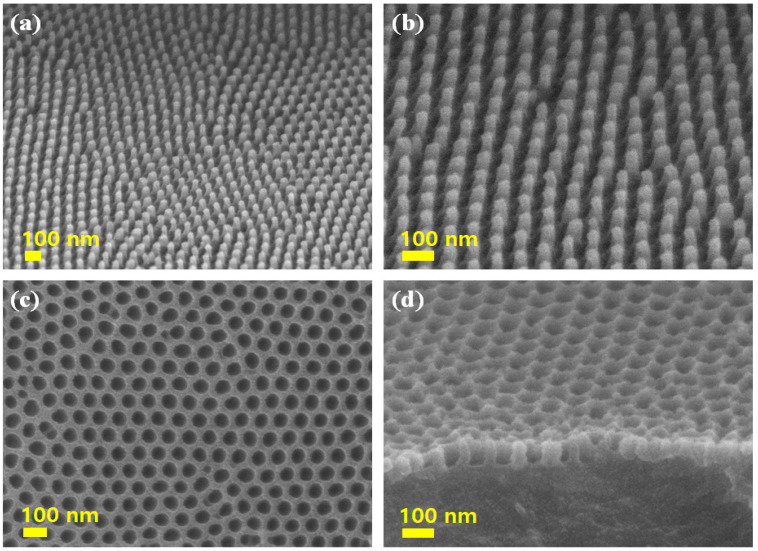
FE-SEM images of (**a**,**b**) embossed and (**c**,**d**) concaved nanostructure using UV-curable polyacrylate urethane resins on a PET substrate after UV irradiation for 30 s, subsequent removal of the AAO template using a fluorine demolding agent, and further UV-curing for 30 s for. The scale bars are 100 nm.

**Figure 4 polymers-13-03333-f004:**
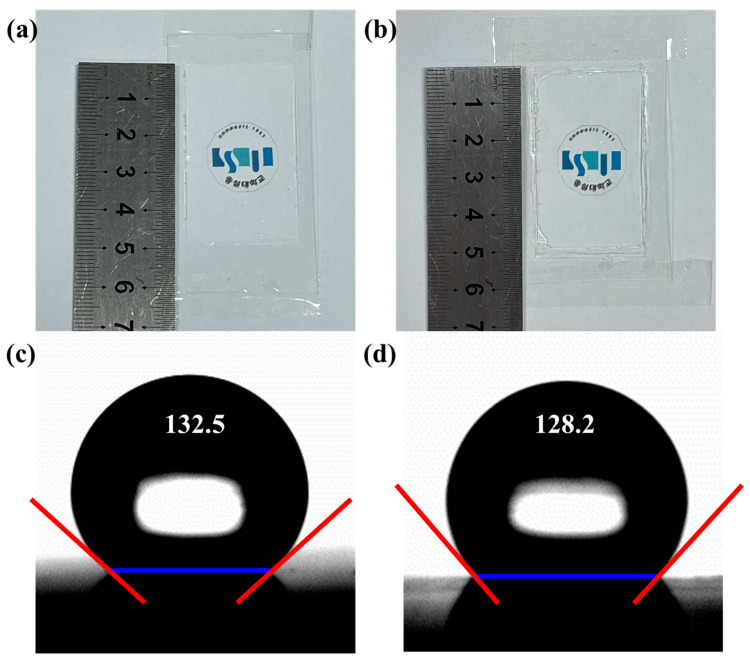
Photos of anti-reflective transparent coating for (**a**) embossed and (**b**) concaved polymer mold. Water contact measurements on (**c**) embossed and (**d**) concaved polymer mold surfaces, to yield the angles at 132.5° and 128.2°, respectively.

**Figure 5 polymers-13-03333-f005:**
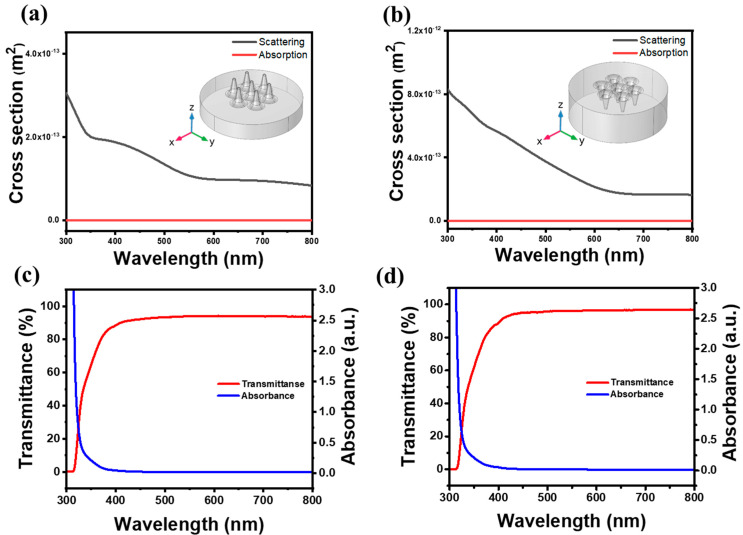
(**a**) Embossed and (**b**) concaved polymeric nanostructures in their absorption and scattering curves calculated by the COMSOL multiphysics software. (**c**) Embossed and (**d**) concaved polymer nanostructures in their transmission and absorption measurements of the nanostructure polymer as illustrated in the SEM images of Figure 4 showing almost perfect transmission between 450 and 800 nm.

**Table 1 polymers-13-03333-t001:** Measured values of contact angles for the embossed and concaved polymeric mold surfaces. The mean and standard deviations were obtained by 10 independent measurements with the units in degree.

	Replica Embossed Mold from Master AAO	Concaved Mold from Embossed Polymer Mold
Mean	132.5	128.2
Standard deviations	2.7	0.9

**Table 2 polymers-13-03333-t002:** Optical transmittance and absorbance values of the embossed polymer replica molds. Ten independent measurements yielded the mean and standard deviation values.

	Wavelength				
	400 nm	500 nm	600 nm	700 nm	800 nm
Transmittance (%)	88.6 (±1.0)	93.4 (±0.6)	94.2 (±0.5)	94.0 (±0.5)	93.9 (±0.5)
Absorbance (a.u.)	0.044 (±0.014)	0.018 (±0.012)	0.017 (±0.011)	0.017 (±0.012)	0.016 (±0.012)

**Table 3 polymers-13-03333-t003:** Optical transmittance and absorbance values of the engraved polymer replica molds obtained from the embossed polymer replica molds. Ten independent measurements yielded the mean and standard deviation values.

	Wavelength				
	400 nm	500 nm	600 nm	700 nm	800 nm
Transmittance (%)	88.9 (±1.0)	95.8 (±0.8)	96.4 (±0.7)	96.8 (±0.6)	96.9 (±0.5)
Absorbance (a.u.)	0.054 (±0.0048)	0.021 (±0.0044)	0.017 (±0.0027)	0.014 (±0.0014)	0.013 (±0.001)

## Data Availability

Data sharing not applicable.

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
