# Peer review of "Nanopatterned Polymer Molds Using Anodized Aluminum Templates for Anti-Reflective Coatings"

_polymers, 2021, doi:10.3390/polym13193333_

Round 1

Reviewer 1 Report

This work presented simple but interesting technology for anti-reflective surfaces preparation. nanotextured polymeric surfaces exhibited good light absorption and superhydrophobicity. The paper looks good but needs some final polishing. Some Figures are not of high quality. Also please see Fig 3b. I don't think that it is possible to evaluate a contact angle from this image. Please provide standard deviations for contact angle and other data.

Author Response

Reply to Reviewer #1's Comments

The authors deeply appreciate the Reviewer #1 for giving helpful comments.

Comment. This work presented simple but interesting technology for anti-reflective surfaces preparation. Nanotextured polymeric surfaces exhibited good light absorption and superhydrophobicity. The paper looks good but needs some final polishing. Some Figures are not of high quality. Also please see Fig 3b. I don't think that it is possible to evaluate a contact angle from this image. Please provide standard deviations for contact angle and other data.

→According to Reviewer #1’s comments, the manuscript is revised as follows.

1)Figures are improved with better quality including Figure 3b in the previous manuscript (Figure 4 in the revised manuscript.

2)The average and standard deviations of the contact angle measurement values are newly added at Table 1 of the revised manuscript.

Finally, the authors would like to thank Reviewer #1 for giving positive comments. Thank you.

Reviewer 2 Report

This manuscript uses anodized alumina (AAO) template method fabricate moth-eye structure on PET film. Moth-eye structure is prepared by UV-curable polyurethane acrylate. However, the AAO template method is very mature and less innovative. The materials used in the fabrication process are also common chemical materials and do not engaging the readers. The manuscript only introduces the antireflection function of the perfect structure. As for the bionic prototype of the manuscript and antireflection principle of structure, the manuscript doesn’t mention the whole article. In the bibliography of references, 18th reference named “Large-scale bio-inspired flexible antireflective film with scale-insensitivity arrays”, AAO templates were used to fabricate AR films. A variety of templates were used to prepare AR films in 18th reference, and the characterization methods are much better than this manuscript. Taking the above points into consideration, this manuscript is not suitable for acceptance at the present stage.

  1. Why the moth-eye structure is chosen to be the template for fabricate AR films?
  2. In previous researches, anodic alumina can produce different nano-morphologies. What are the advantages of shapes in this manuscript compared to others?
  3. In many nano-structures studies, the aspect ratio is an important factor for the antireflection effect. Is it possible to explain the differences of these 4 anodic alumina templates from the perspective of aspect ratio?
  4. In line 57, “to the best of our knowledge” is not suitable enough in the manuscript as the template method was mature in the last few years.
  5. In line 141, it is mentioned in the manuscript that “as large as a few centimeters”, it is suggested to tell the specific size of the AR film in the fabrication process. It could be better to exhibit the image of film in the manuscript.
  6. In line 154, it is mentioned “We have found …”, what is the reason why other templates cannot fabricate stable polymer mold shape. It is suggested to clarify in the manuscript.
  7. In line 165, what is the surface-treatment polymer?
  8. The water contact angle of AAO and polymer surface is 127° and 120°. However, the contact angle does not indicate that films have self-cleaning and self-curing behavior. More relevant experiments are recommended.
  9. In line 194, it is mentioned that this kind film is supposed to be applied in outdoor sensor devices. Can make a small demonstration to show its potential application?
  10. In Figure 5c, the transmittance curves are almost 100%, it is difficult to achieve this high transmittance under normal condition. Even the thin glass slide could not reach 100% transmittance. The experimental condition should be added in the manuscript.

Author Response

Reply to Reviewer #2's Comments

The authors deeply appreciate the Reviewer #2 for giving helpful comments.

Comment. This manuscript uses anodized alumina (AAO) template method fabricate moth-eye structure on PET film. Moth-eye structure is prepared by UV-curable polyurethane acrylate. However, the AAO template method is very mature and less innovative. The materials used in the fabrication process are also common chemical materials and do not engaging the readers. The manuscript only introduces the antireflection function of the perfect structure. As for the bionic prototype of the manuscript and antireflection principle of structure, the manuscript doesn’t mention the whole article. In the bibliography of references, 18th reference named “Large-scale bio-inspired flexible antireflective film with scale-insensitivity arrays”, AAO templates were used to fabricate AR films. A variety of templates were used to prepare AR films in 18th reference, and the characterization methods are much better than this manuscript. Taking the above points into consideration, this manuscript is not suitable for acceptance at the present stage.

→ We attempted to address the issues raised by the Reviewer #2 as follows:

  1. i) The following sentences are added at the 44-48th lines from page 1 through page 2 in the revised manuscript.

“The bionic prototypes and antireflection principles inspired by nature are discussed in the recent studies. Hexagonal nipple-array of subwavelength conical nanostructures with the diameter of ca. 100 nm and can provide broadband anti-reflectivity to enhance photon collection capability. The sizes and shapes appeared to affect the reflectivity performance [4,5].”

  1. ii) The following references are added to address the issue of the bionic prototypes and antireflection principles in the revised manuscript.

[4] Motamedi, M.; Warkiani, M. E.; Taylor, R. A. Transparent surfaces inspired by nature. Adv. Opt. Mater. 2018, 1, 1800091.

[5] Raut, H. K.; Ganesh, V. A.; Nairb, A. S.; Ramakrishna, S. Anti-reflective coatings: A critical, in-depth review. Energy Environ. Sci. 2011, 4, 3779–3804.

iii)The characterization methods are improved in the revised manuscript.

We improved the characterization methods of contact angles and optical measurements by providing the averages and standard deviations as newly listed in Tables 1-3 along with Figures 4 and 5 by repeating 10 times.

Point to Point Responses

  1. Why the moth-eye structure is chosen to be the template for fabricate AR films?

→The moth eye nanostructures are chosen due to their broad band antireflection properties (refs. 4 -8) and easy preparation using AAO templates, as recently reported by the study (Yoo, Y. J.; Kim, Y. J.;  Kim, S-Y.; Lee, J. H.; Kim, K.; Ko, J. H.; Lee, J. W.; Lee, B. H.; Song, Y. M. Mechanically robust antireflective moth-eye structures with a tailored coating of dielectric materials. Opt. Mater. Express 2019, 9, 4178–4186.

  1. In previous researches, anodic alumina can produce different nano-morphologies. What are the advantages of shapes in this manuscript compared to others?

→The concaved structures obtained from the polymer molds are found to have an advantage of higher optical transmission properties than those of the conventional embossed ones produced from the AAO master molds as in the previous reports as stated at the 211-221st lines of Page 8 in the revised manuscript.

  1. In many nano-structures studies, the aspect ratio is an important factor for the antireflection effect. Is it possible to explain the differences of these 4 anodic alumina templates from the perspective of aspect ratio?

→ Regarding the antireflective properties, the sentence of “The sizes and shapes appeared to affect the reflectivity performance [6-8]” was newly added at the 47-48th lines of page 2 in the revised manuscript.

4 . In line 57, “to the best of our knowledge” is not suitable enough in the manuscript as the template method was mature in the last few years.

→The problematic phrases are deleted in the revised manuscript. The following sentence is added at the 68-70th lines of page 2 in the revised manuscript.

“Based on the previous routine procedures, there is a practical need to develop the fabrication of various polymer-based moth-eye nanostructures”

  1. In line 141, it is mentioned in the manuscript that “as large as a few centimeters”, it is suggested to tell the specific size of the AR film in the fabrication process. It could be better to exhibit the image of film in the manuscript.

→Photos to show the dimension of the prepared AR films are newly added in Figure 5(a) and (b) in the revised manuscript.

  1. In line 154, it is mentioned “We have found …”, what is the reason why other templates cannot fabricate stable polymer mold shape. It is suggested to clarify in the manuscript.

→The demolding properties in the polymer molds from the different anodic alumina templates are presumably due to the van der Waals interactions as mentioned at the 177-178th lines of Page 5 in the revised manuscript.

  1. In line 165, what is the surface-treatment polymer?

→Fluorosilane-based solution was used for the surface treatments.

  1. The water contact angle of AAO and polymer surface is 127° and 120°. However, the contact angle does not indicate that films have self-cleaning and self-curing behavior. More relevant experiments are recommended.

→It is expected that the prepared polymer surfaces should have a certain self-cleaning and self-curing behaviors (Maghsoudi, K.; Vazirinasab, E.; Momen, G.; Jafari, R. Advances in the fabrication of superhydrophobic polymeric surfaces by polymer molding processes. Ind. Eng. Chem. Res. 2020, 59, 9343–9363).

  1. In line 194, it is mentioned that this kind film is supposed to be applied in outdoor sensor devices. Can make a small demonstration to show its potential application?

→ The sizes and good transparent properties may be suitable for the coating of LiDAR windows using 905 nm near infrared laser light for its potential application.

  1. In Figure 5c, the transmittance curves are almost 100%, it is difficult to achieve this high transmittance under normal condition. Even the thin glass slide could not reach 100% transmittance. The experimental condition should be added in the manuscript.

→The transmittance values are corrected after the repetitive measurements as demonstrated in Figure 5(c) and (d) along with Tables 2 and 3.

Finally, the authors would like to thank Reviewer #2 for giving valuable comments. Thank you.

Reviewer 3 Report

Please consider reviewing the abstract. Most importantly please highlight the exact (detailed) main findings of this work, the ones in the abstract now are somewhat generic and do not provide clear idea of what was found from this work.

Please check the paper for English editing and typos.

Before the end of the introduction the authors should attempt to answer the following question: what the research gap is you found from the literature review similar or related to your work, mention it as it will strengthen the quality of your article.

The introduction should be improved, please provide more critical discussion instead of just telling us generic information from the literature, it would be better if the authors can critically compare the past literature against each other and make their conclusions about their work, also expand the introduction, it reads short and with limited information as it is now.

The first paragraph of the introduction there is somewhat an abuse of citations, 11 citations just to tell the readers about the applications of nanostructures. This is not acceptable please improve this paragraph further.

Please avoid writing small paragraphs of 4 lines or less. There is quiet few in the manuscript which can be combined into larger paragraphs to maintain good flow and readability of the article.

Materials and method section lacks any graphical images. Please add graphs and images of test equipment, test setup, fabricated samples and so on. This is an experimental study, and it is very important to provide graphical information for the readers about this part.

Figure 1 please add numbers beside the scale bar in each of the images.

Also figure 1 seems to belong to the Materials and method section so please move it up there.

Line 128-148 this part belongs to the materials and method section, why the authors add in the results and discussion section?

Line 153, please explain this observation further and support with references.

Line 162 “well manufactured” what does the authors means by this, please explain further. How do you know it was well manufactured, is there something which you measure and compare some referenced values to say this is well or poorly manufactured?

Add scale bar to figure 3 (scale bar with numbers)

I am not sure what are we exactly looking at in Figure 4, it is not even clear if they have same resolution and same scale. What are the authors trying to show/tell the readers here?

Line 184 this is an abuse of citation, adding a bulk of citations for one sentence is discouraged unless each reference is given full credit somewhere in the article.

Please avoid using we, our in the article and use proper writing format. Please check this issue everywhere in the manuscript.

More details are needed about the COMSOL setup/simulation process.

All the graphs in this article need some improved, make their size same, reduce the font size of (a) (b) and (c) if any in the figure.

Conclusion is weak and doesn’t reflect what was done in this work, also it is recommended to use bullet points for the most important findings from this study.

The authors are encouraged to include more detailed discussion section and critically discuss the observations from this investigation with existing literature. There is literally almost no discussion in this article, only less than one paragraph!

Usually i dont comment about this but there are 11 authors for 8 pages of work which is somewhat unrealistic in my opinion for the size of the work done here. 

Author Response

Reply to Reviewer #3's Comments

The authors deeply appreciate the Reviewer #3 for giving helpful comments and corrected unclear expressions and errors in the revised manuscript. According to Reviewer #3’s comments, we revised the manuscript as follows:

1) Please consider reviewing the abstract. Most importantly please highlight the exact (detailed) main findings of this work, the ones in the abstract now are somewhat generic and do not provide clear idea of what was found from this work.

→According to Reviewer #3’s comments, the following sentences are added in the revised manuscript.

“Both embossed and concaved polymer moth-eye nanostructures were successfully mounted on a PET substrate. Embossed polymer replica molds were prepared using the AAO master templates in combination with an imprinting process…The polymer molds with the depths of 300 and 500 nm revealed the joined structures in their apexes. In addition, a dip-imprinting process of the polymer layers was implemented to yield a concaved mold by assembly on the surface of the 100 nm embossed polymer mold substrate. Considering that the embossed structures may be crumbled due to their protuberant shapes, the concaved geometries can have an advantage of stability in a certain application concerning physical degradation along with a higher transmission by ~2%, despite somewhat nonuniform structure”

2)Please check the paper for English editing and typos.

→ We attempted to improve our manuscript by correcting some mistakes in the content of manuscript by consulting professional English editors. We consulted a professional English editor with the attached certificate.

3)Before the end of the introduction the authors should attempt to answer the following question: what the research gap is you found from the literature review similar or related to your work, mention it as it will strengthen the quality of your article.

→We attempted to manifest what we have found in the present work in comparison with the previous related reports as summarized with the bullet points in the conclusion parts.

4)The introduction should be improved, please provide more critical discussion instead of just telling us generic information from the literature, it would be better if the authors can critically compare the past literature against each other and make their conclusions about their work, also expand the introduction, it reads short and with limited information as it is now.

→We attempted to improve the introduction by comparing the past literature.

5)The first paragraph of the introduction there is somewhat an abuse of citations, 11 citations just to tell the readers about the applications of nanostructures. This is not acceptable please improve this paragraph further.

→Somewhat overused citations regarding the nanostructures are omitted in the revised manuscript. The citations of [2,4,6,7,8] are deleted for better presentation.

6)Please avoid writing small paragraphs of 4 lines or less. There is quiet few in the manuscript which can be combined into larger paragraphs to maintain good flow and readability of the article.

→Small paragraphs with less than four lines are omitted in the revised manuscript. 

7)Materials and method section lacks any graphical images. Please add graphs and images of test equipment, test setup, fabricated samples and so on. This is an experimental study, and it is very important to provide graphical information for the readers about this part.

→Figure 1 is moved to the Materials and Method section after revising the schemes by including the test equipment.   

8)Figure 1 please add numbers beside the scale bar in each of the images.

→Numbers are added above the scale bars in Figure 1.

9)Also figure 1 seems to belong to the Materials and method section so please move it up there.

→Figure 1 is moved to the Materials and Method section.

10)Line 128-148 this part belongs to the materials and method section, why the authors add in the results and discussion section?

→Line 128-148 of “Figure 1 illustrates…, thus generating a replica moth-eye nanostructure” is moved to the materials and method section.

11)Line 153: “The shapes and sizes of imprinted polymer-based moth-eye-like nanostructure ar-rays were found to depend on the structures of nanopores arrays of AAO template molds”, please explain this observation further and support with references.

→The following references are newly added in the revised manuscript.

[6] Stavenga, D.G.; Foletti, S.; Palasantzas, G.; Arikawa, K. Light on the moth-eye corneal nipple array of butterflies. Proc. R. Soc. B 2006, 273, 661–667.

[7] Córdova-Castro, R. M.; Krasavin, A. V.; Nasir, M. E.; Zayats, A. V.; Dickson, W. Nanocone-based plasmonic metamaterials. Nanotechnology 2019, 30, 055301.

[8] Zhangyang, X.; Liu, L.; Lv, Z.; Lu, F.; Tian. The effect of geometry parameters on light harvesting performance of GaN nanostructure arrays—a numerical investigation and simulation. Mater. Res. Express 2020, 7, 015009.

12)Line 162 “well manufactured” what does the authors means by this, please explain further. How do you know it was well manufactured, is there something which you measure and compare some referenced values to say this is well or poorly manufactured?

→“well manufactured” is changed to “well-controlled” in the revised manuscript.

13)Add scale bar to figure 3 (scale bar with numbers)

→Scale bars are included in Figure 3.

14)I am not sure what are we exactly looking at in Figure 4, it is not even clear if they have same resolution and same scale. What are the authors trying to show/tell the readers here?

→Figure 4 is presented newly along with both the embossed and concaved structures in the revised manuscript.

15)Line 184: “Polymer moth-eye nanostructures have been found to exhibit anti-reflective properties according to the previous works [3,10,14,19,33,34]” this is an abuse of citation, adding a bulk of citations for one sentence is discouraged unless each reference is given full credit somewhere in the article.

→Overused citations in a sentence were changed to [33,34] in the revised manuscript.

16)Please avoid using we, our in the article and use proper writing format. Please check this issue everywhere in the manuscript.

→Either “we” or “our” is omitted in the revised manuscript.

17) More details are needed about the COMSOL setup/simulation process.

→The following sentences were added in the revised manuscript.

“COMSOL Multiphysics Version 5.3a package is used to simulate the absorption and scattering spectra of Moth Eye Structure (MES) with incident light. The geometric parameters of the nanostructure are realized by the ‘parameter’ function in ‘global definition’, which can easily change the parameter settings of the entire model. The one single hexagonal structure of MES with 6 tips was chosen to represent the MES array. The sizes of MES were chosen following the table 1, the length and the tip radius were 100 nm and 45 nm, respectively, and the thickness of substrate was 100nm. The material model contains the refractive index of MES and air. The model assumes that the real part of the refractive index of air is 1 and the imaginary part of the refractive index is 0. The refractive index of MES was chosen as 1.5. The length The MES substrate was introduced into the middle of an air spherical domain (f= 2 μm) to avoid reflection by a perfectly matched layer boundary condition. The incident light wavelength range of this simulation model is 400 nm to 2000 nm, so the maximum cell size is 65 nm and the minimum cell size is 2 nm.”

18)All the graphs in this article need some improved, make their size same, reduce the font size of (a) (b) and (c) if any in the figure.

→We attempted to improve the graphs including the reduction of the font sizes of (a), (b), and (c) in the revised manuscript.

19) Conclusion is weak and doesn’t reflect what was done in this work, also it is recommended to use bullet points for the most important findings from this study.

→We attempted to improve the conclusion part by using bullet points.

  • Both embossed and engraved nanopatterned polymer mold structures were tested for potential applications in transparent coating.
  • Only an AAO master mold in the depth of 100 nm is found to be eligible to produce a stable embossed polymer mold shape.
  • The neighboring apexes in nanocones appeared to be amalgamated for the 300 and 500 nm templates in depth.
  • Contact angle measurements of embossed and concaved polymer surfaces were estimated to be 132.5 (±2.7)° and 128.2 (±0.9)°, respectively
  • Concaved molds showed approximately 2% higher transmittance values than those of the embossed ones.

20)The authors are encouraged to include more detailed discussion section and critically discuss the observations from this investigation with existing literature. There is literally almost no discussion in this article, only less than one paragraph!

→We attempted to improve the discussion section by adding the observed data of contact angle and optical transmission values for both embossed and concaved polymer molds.

21)Usually i dont comment about this but there are 11 authors for 8 pages of work which is somewhat unrealistic in my opinion for the size of the work done here.

→We attempted to explain the authors’ contribution in more details. Due to the potential coating applications from the commercial products, a number of students have been involved in the present work supported by various funding agencies.

Conceptualization, S.H. Lim and N.H. Ly; methodology, J.A Lee and J.E. Kim; UV imprint mold investigation, S-W. La and V.T. Huong; computed COMSOL simulation, supporting polymer resin experiments, N.T. Ho and T-G. Tran; writing—original draft preparation, N.H. Ly; writing—review and editing, S.J. Son and S.M. Noh; commercial application, visualization and supervision, S.J. Son and S-W. Joo. All authors have agreed to the published version of the manuscript.

Finally, the authors would like to thank Reviewer #2 for giving valuable comments. Thank you.

Round 2

Reviewer 3 Report

All questions answered and paper can be accepted